# MemoryFormer: Minimize Transformer Computation by Removing Fully-Connected Layers

**Ning Ding**[1*], **Yehui Tang**[2*], **Haochen Qin**[2], **Zhenli Zhou**[1], **Chao Xu**[1], **Lin Li**[3],
**Kai Han**[2], **Heng Liao**[3], **Yunhe Wang**[2†]

[1] State Key Lab of General AI, School of Intelligence Science and Technology, Peking University
[2] Huawei Noah's Ark Lab.      [3] Huawei HiSilicon
dingning@stu.pku.edu.cn      xuchao@cis.pku.edu.cn
{yehui.tang, yunhe.wang}@huawei.com

## Abstract

In order to reduce the computational complexity of large language models, great efforts have been made to to improve the efficiency of transformer models such as linear attention and flash-attention. However, the model size and corresponding computational complexity are constantly scaled up in pursuit of higher performance. In this work, we present **MemoryFormer**, a novel transformer architecture which significantly reduces the computational complexity (FLOPs) from a new perspective. We eliminate nearly all the computations of the transformer model except for the necessary computation required by the multi-head attention operation. This is made possible by utilizing an alternative method for feature transformation to replace the linear projection of fully-connected layers. Specifically, we first construct a group of in-memory lookup tables that store a large amount of discrete vectors to replace the weight matrix used in linear projection. We then use a hash algorithm to retrieve a correlated subset of vectors dynamically based on the input embedding. The retrieved vectors combined together will form the output embedding, which provides an estimation of the result of matrix multiplication operation in a fully-connected layer. Compared to conducting matrix multiplication, retrieving data blocks from memory is a much cheaper operation which requires little computations. We train MemoryFormer from scratch and conduct extensive experiments on various benchmarks to demonstrate the effectiveness of the proposed model.

## 1 Introduction

The Transformer model has made magnificent achievement in deep learning community since it was made public. The Transformer not only successfully leads the revolution in the field of natural language processing due to its excellent performance, but also motivates the innovation in model architecture in other fields such as computer vision and speech recognition. Recently, large language models (LLMs), transformers that are extremely scaled up in size, have drawn remarkable attention of both researchers and non-researchers across the globe. The unprecedented emergent abilities that LLMs demonstrate attract an increasing number of investments and researches, which point out a potential pathway for the artificial general intelligence.

However, what comes along with the scaling lay is not only more intelligence, but also greater consumption of computing resources. The ever-increasing computational complexity is currently the main obstacle hindering the application and popularization of LLMs. In response to this situation,

---

[*]Equal Contribution.    [†]Corresponding Author.

38th Conference on Neural Information Processing Systems (NeurIPS 2024).

great efforts have been made towards optimizing the architecture of transformer model by the research community. Some works using traditional methods such as model pruning and weight quantization are able to lower the computational complexity of LLMs to some degree. Another line of works that are specialized for transformer re-design the self-attention mechanism, which is the key to sequence modeling. They use sliding-windows or kernel function to reduce the complexity from quadratic to sub-quadratic or even linear with respect to the sequence length, while maintaining a comparable performance.

According to our observation, in most application scenarios, only a small proportion of the computational complexity comes from the multi-head attention (MHA) operation, while the majority of the computation comes from the fully-connected (FC) layers in the transformer model. Specifically, given a standard transformer model with the hidden size being $d$ and the length of input sequence being $s$, the amount of floating-point computation of the MHA operation is $2s^2d$, and the computation in all the FC layers is $12sd^2$. The computation required by the MHA becomes dominant only when $s > 6d$. That's to say, for an LLM with hidden size $d = 4096$, the sequence length $s$ needs to be larger than 24K.

Another observation we have is that, at present, the inference stage of deep neural network relies on the parallel-computing cores of the graphics processing unit (GPU), while the CPU and the random-access memory (RAM) resources in the computer system are left almost unused, despite the fact that the size of the RAM easily reaches terabytes and the CPU has hundreds of cores (*e.g.* NVIDIA DGX A100 has 2TB RAM and 128 CPU cores [21]). What's more, CPU manufacturers have started developing tensor core to accelerate parallel-computing, which might make the low-latency CPU inference feasible in the future.

Based on the above observations, we present a novel transformer architecture in this work, named **MemoryFormer**, to minimize the required computational complexity from a new perspective. Instead of the fully-connected layers used in a standard transformer, MemoryFormer uses the Memory Layer to process the feature vector (the embedding of token). Specifically, the Memory Layer contains a group of in-memory hash tables which store a large amount of discrete vectors. It uses locality-sensitive hashing (LSH) algorithm to retrieve a subset of vectors from the hash tables that are correlated with the input token embedding. These retrieved vectors are then aggregated with different weights to form the output of the Memory Layer. The result of hashing-and-aggregating operation performed by the Memory Layer provides an estimation of the matrix multiplication of the fully-connected layer. We also design a method to make all the vectors stored in hash tables learnable via the back-propagation of gradients. Therefore, the MemoryFormercan be trained from scratch in an end-to-end manner.

Since the amount of computation produced by the hash operation is negligible, the absolute majority of the computations now results from the matrix-multiplications within the multi-head attention. Figure 1 demonstrates the comparison of the computational complexity of one block between the proposed MemoryFormer (red) and the baseline transformer (blue). The MemoryFormer block only requires ∼19% of the FLOPs compared with the baseline transformer block when sequence length $s = 2048$ and hidden size $d = 2048$. The effect of FLOPs-reduction is going to be more significant as the model size is scaled up. Replacing all fully-connected layers with Memory Layers allows us to trade the memory resources for less computational complexity.

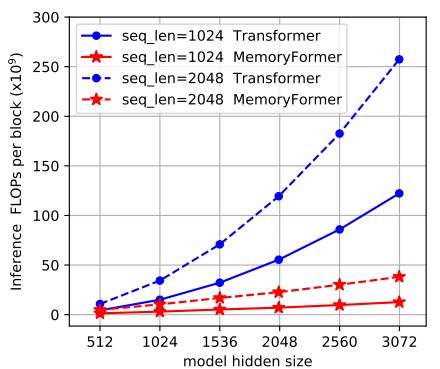

Figure 1: FLOPs with different model hidden size and sequence lengths.

This work not only proposes a new FLOPs-reduction strategy that is different from existing approaches, but also provides guiding significance for the hardware design (*e.g.* bigger bus width and higher cache hit rate) of the next-generation parallel-computing platform. We train a series of MemoryFormers of different sizes and validate the effectiveness of these models on multiple public benchmarks. The experiment results show that our method can achieve performance comparable to the baseline Transformer with significantly less computation.

## 2 MemoryFormer

### 2.1 Background

In the standard Transformer model, there are two main types of operations to perform feature transformation on the sequence of token embeddings. One is the multi-head attention (MHA), the most important operation in a transformer block which captures the long-range inter-relationships among different tokens within the sequence. The other is the ubiquitous fully-connected (FC) layer that performs linear projection on each token in the sequence separately. In addition to the projections of $\{\mathbf{W}_Q, \mathbf{W}_K, \mathbf{W}_V\}$ prior to the MHA, the feed-forward network (FFN) is also composed of multiple FC layers.

Let $\mathbf{x} \in \mathbb{R}^d$ be a row vector representing any token embedding, a fully-connected layer parameterized by weight matrix $\mathbf{W} \in \mathbb{R}^{d \times h}$ applies a linear projection to $\mathbf{x}$ formulated as $\mathbf{y} = \mathbf{xW}$, where $\mathbf{y} \in \mathbb{R}^h$ is the output token embedding. For a sequence composed of $s$ tokens, this would become a matrix multiplication in Eq. (1) with computational complexity of $\mathcal{O}(sdh)$.

$$\mathbf{Y} = \mathbf{XW}, \quad \mathbf{X} \in \mathbb{R}^{s \times d}, \ \mathbf{y} \in \mathbb{R}^{s \times h}. \tag{1}$$

In finite-dimensional vector space, the fully-connected layer is a continuous linear operator, which means, for two adjacent input feature vector $\mathbf{x}_1$ and $\mathbf{x}_2$, given the same weight matrix $\mathbf{W}$, the projected vectors $\mathbf{y}_1 = \mathbf{x}_1 \mathbf{W}$ and $\mathbf{y}_2 = \mathbf{x}_2 \mathbf{W}$ are most likely to be similar as well. In this work, we want to find an alternative mapping function which has much less computational complexity than $\mathcal{O}(sdh)$ yet generally in accord with the properties of the linear projection. If this is achieved, we can use this alternative method to replace all the FC layers to perform feature transformation while reducing computation.

In this paper, we use normal-font lowercase letters to denote scalars (*e.g.* $d$ and $h$), bold-font lowercase letters to denote vectors (*e.g.* $\mathbf{x}$ and $\mathbf{y}$), and bold-font uppercase letters to denote matrices (*e.g.* $\mathbf{W}$). We use the notation $[\cdot]_i$ to represent the $i$-th row of a matrix, and also use $[\cdot]_i$ to represent the $i$-th entry of a vector.

### 2.2 Compute-less Locality-Sensitive Hashing

Unlike ordinary hash functions that are designed to avoid collisions for different encoded items, the aim of locality-sensitive hashing (LSH) is to map similar items to the same hash bucket (memory location) in the hash table. For example, in the text-to-image search system, several different descriptions for the same image are expected to have an identical hash code after encoded by the LSH function, and thus retrieve the same image.

We decide to apply LSH function in the embedding space to encode any input feature vectors. Assuming $\mathbf{x}$ is hashed to a specific bucket that stores a vector $\hat{\mathbf{y}}$ by the LSH function, then the hashing result for some adjacent neighbor vectors of $\mathbf{x}$ will correspond to the same hash bucket and retrieve $\hat{\mathbf{y}}$ as well. If the value $\hat{\mathbf{y}}$ in the hash table is an approximation of the result of linear operation $\mathbf{y} = \mathbf{xW}$ for the input vector $\mathbf{x}$, we can use this method, that is to find an estimated result in a hash table, to replace the fully-connected layer, while only using the FLOPs of the hashing operation.

Firstly, we construct an in-memory hash table parameterized by the matrix $\mathbf{T} \in \mathbb{R}^{2^d \times h}$ which stores $2^d$ vectors $[\mathbf{T}]_i \in \mathbb{R}^h$. Traditional LSH functions, such as Hyperplane-LSH [5], incorporate multiple linear projections to generate the hash code for table indexing. To avoid any unnecessary computations, we utilize a much simpler LSH function to generate the hash code $h(\mathbf{x})$. Specifically, the process of hashing and retrieving the result from table $\mathbf{T}$ is formulated as:

$$h(\mathbf{x}) = \text{integer}(\text{sign}(\mathbf{x})), \tag{2}$$

$$\text{sign}([\mathbf{x}]_i) = \begin{cases} -1, & \text{if } [\mathbf{x}]_i < 0, \\ 1, & \text{if } [\mathbf{x}]_i \geq 0, \end{cases} \tag{3}$$

$$\text{integer}(\mathbf{s}) = \sum_{i=0}^{d-1} \frac{[\mathbf{s}]_i + 1}{2} \cdot 2^i, \tag{4}$$

$$\hat{\mathbf{y}} = [\mathbf{T}]_{h(\mathbf{x})}, \tag{5}$$

where $\mathbf{x} \in \mathbb{R}^d$ is the input vector, $\mathbf{s} = \text{sign}(\mathbf{x}) \in \{-1, 1\}^d$ is the corresponding binary representation (hash code), integer($\cdot$) function converts the binary representation $\mathbf{s}$ to the corresponding non-negative integer used as the index number of the hash table, $h(\mathbf{x}) \in \{0, 1, 2, \cdots, 2^d - 1\}$. Notably, the space complexity of such a hash table is $\mathcal{O}(2^d h)$. The required memory space would be $\sim 10^{145}$ terabytes (TB) when using float16 datatype with $d = h = 512$, which is impractical for any modern computer system.

To tackle this problem, we propose to evenly split $\mathbf{x} \in \mathbb{R}^d$ into $K$ non-overlapping chunks and handle them separately:

$$\mathbf{z}_k = \text{split}(\mathbf{x}, \text{num\_chunk} = K), \quad k = 1, 2, \cdots, K, \tag{6}$$

where $\mathbf{z}_k \in \mathbb{R}^\tau$, $\tau = \frac{d}{K}$, and $d$ is evenly divisible by $K$. We then set up a hash table $\mathbf{T}_k \in \mathbb{R}^{2^\tau \times h}$ for each sub-vector $\mathbf{z}_k$, respectively. Therefore, the output result is

$$\hat{\mathbf{y}} = \sum_{k=1}^{K} [\mathbf{T}_k]_{h(\mathbf{z}_k)}, \tag{7}$$

Since $\mathbf{z}_k$ has a smaller bit width after binarization and thus the corresponding hash table $\mathbf{T}_k$ would comsume less memory space for storage. The space complexity of Eq. (7) is $\mathcal{O}(K2^\tau h)$. When $d = h = 512, \tau = 8, K = 64$ and data type is float16, the storage required by all $K$ hash tables is $\sim 16$ MegaBytes(MB). Figure 2 is a simple demonstration of the proposed locality-sensitive hashing.

## 2.3 Memory Layer

So far, the above-mentioned formulation is able to simulate the forward pass of fully-connected layer. And the values store in the hash tables can be updated via back-propagation. The derivative of the loss function $L$ with respect to the hash table is $\frac{\partial L}{\partial [\mathbf{T}_k]_{h(\mathbf{z}_k)}} = \frac{\partial L}{\partial \hat{\mathbf{y}}} \frac{\partial \hat{\mathbf{y}}}{\partial [\mathbf{T}_k]_{h(\mathbf{z}_k)}}$. However, the input vector $\mathbf{x}$ is unable to have gradient since it's hashed to multiple integers $h(\mathbf{z}_k)$ used as the index number for retrieval, which is a non-differentiable operation. If we can reformulate Eq. (7) as $\hat{\mathbf{y}} = \sum_{k=1}^{K} p(\mathbf{z}_k) \cdot [\mathbf{T}_k]_{h(\mathbf{z}_k)}$ to add a coefficient $p(\mathbf{z}_k)$ to weight each retrieved item, where $p(\mathbf{z}_k)$ is a function of the variable $\mathbf{z}_k$, the gradients can be back-propagated to the input $\mathbf{x}$ via $[\mathbf{T}_k]_{h(\mathbf{z}_k)}$.

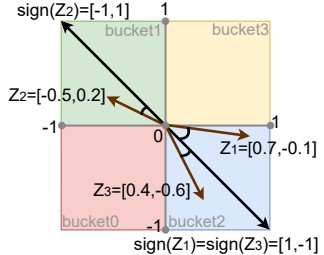

Figure 2: A demonstration with $\tau = 2$ and $K = 3$, where $\mathbf{z}_1$ is hashed to the bucket2 of $\mathbf{T}_1$, $\mathbf{z}_2$ is hashed to the bucket1 of $\mathbf{T}_2$, $\mathbf{z}_3$ is hashed to the bucket2 of $\mathbf{T}_3$.

As we can observe from Figure 2, many sub-vectors with various directions and amplitudes can still be hashed to the same bucket as long as their signs are identical, but the angle between the bucket's representative binary vector (each entry is either 1 or -1) and these sub-vectors are different, which is defined by the cosine value $\cos(\mathbf{z}_k, \text{sign}(\mathbf{z}_k))$. We use a scaled cosine similarity, which takes into account both the direction and amplitude of $\mathbf{z}_k$, to measure the relevance between $\mathbf{z}_k$ and its corresponding hash bucket $h(\mathbf{z}_k)$:

$$\text{sim}(\mathbf{z}_k, h(\mathbf{z}_k)) = \|\mathbf{z}_k\|_2 \cdot \|\text{sign}(\mathbf{z}_k)\|_2 \cdot \cos(\mathbf{z}_k, \text{sign}(\mathbf{z}_k)) = \langle \mathbf{z}_k, \text{sign}(\mathbf{z}_k) \rangle, \tag{8}$$

where $\langle, \rangle$ computes the inner-product of two vectors. Considering all the $2^\tau$ buckets that $\mathbf{z}_k$ is possibly hashed to in the lookup table $\mathbf{T}_k$ simultaneously, we define the probability that $\mathbf{z}_k$ is specifically mapped to the $h(\mathbf{z}_k)$-th hash bucket:

$$p(\mathbf{z}_k) = \frac{exp[\text{sim}(\mathbf{z}_k, h(\mathbf{z}_k))/t\,]}{\sum\limits_{i=0}^{2^\tau - 1} exp[\text{sim}(\mathbf{z}_k, i)/t\,]} = \frac{exp[\langle \mathbf{z}_k, \text{sign}(\mathbf{z}_k) \rangle / t\,]}{\sum\limits_{i=0}^{2^\tau - 1} exp[\langle \mathbf{z}_k, \text{integer}_\tau^{-1}(i) \rangle / t\,]}, \tag{9}$$

where $t$ is the temperature hyper-parameter, $\text{integer}_\tau^{-1}(i) \in \{-1, 1\}^\tau$ is a function that maps an non-negative integer $0 \leq i < 2^\tau$ to the corresponding $\tau$-bit binary representation. Note that $\langle \cdot, \text{integer}_\tau^{-1}(i) \rangle$ operator takes the summation after elementwise selective sign-flipping over any $\tau$-dimensional vector, therefore we have

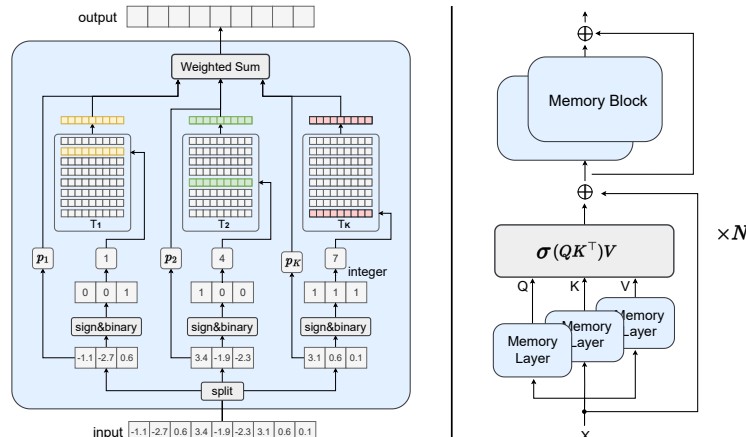

Figure 3: **Left**: The schematic diagram of the Memory Layer. **Right**: One building block of the MemoryFormer.

$$\langle \mathbf{z}_k, \text{sign}(\mathbf{z}_k) \rangle = \sum_{i=0}^{\tau-1} |[\mathbf{z}_k]_i| ,$$

$$\sum_{i=0}^{2^\tau - 1} exp[\langle \mathbf{z}_k, \text{integer}_\tau^{-1}(i) \rangle] = \prod_{i=0}^{\tau-1} [exp([\mathbf{z}_k]_i) + exp(-[\mathbf{z}_k]_i)] , \tag{10}$$

$$p(\mathbf{z}_k) = \frac{exp(\ \sum_{i=0}^{\tau-1} |[\mathbf{z}_k]_i|/t\ )}{\prod_{i=0}^{\tau-1} [exp([\mathbf{z}_k]_i/t\ ) + exp(-[\mathbf{z}_k]_i/t\ )]} = \frac{1}{\prod_{i=0}^{\tau-1} [1 + exp(-2|[\mathbf{z}_k]_i|/t\ )]}.$$

This way, we can formulated the Memory Layer as

$$\mathbf{y} = \sum_{k=1}^{K} p(\mathbf{z}_k) \cdot [\mathbf{T}_k]_{h(\mathbf{z}_k)} = \sum_{k=1}^{K} \frac{[\mathbf{T}_k]_{h(\mathbf{z}_k)}}{\prod_{i=0}^{\tau-1} [1 + exp(-2|[\mathbf{z}_k]_i|/t\ )]}. \tag{11}$$

The left part of Figure 3 illustrates the schematic of the Memory Layer. For a sequence composed of $s$ $d$-dimensional tokens, and the dimensionality of output embeddings is $h$, the computation complexity of a Memory Layer in Eq. (11) is $\mathcal{O}(s(\tau + h)K) \approx \mathcal{O}(\frac{sdh}{\tau})$. This is one order of magnitude smaller than the fully-connected layer which is $\mathcal{O}(sdh)$ when $\tau = 10$.

According to Eq. (11), we can compute the derivative of the loss function $L$ with respect to both the hash tables and the input vector as follows:

$$\frac{\partial L}{\partial [\mathbf{T}_k]_i} = \begin{cases} p(\mathbf{z}_k) \frac{\partial L}{\partial \mathbf{y}}, & \text{if } h(\mathbf{z}_k) = i, \\ 0, & \text{if } h(\mathbf{z}_k) \neq i, \end{cases} \quad i \in \{0, 1, \cdots, 2^\tau - 1\},$$
$$\frac{\partial L}{\partial \mathbf{x}} = \text{concat}(\ [\dots \frac{\partial L}{\partial y} [\mathbf{T}_k]_{h(\mathbf{z}_k)}^\top \frac{\partial p(\mathbf{z}_k)}{\partial \mathbf{z}_k} \dots \text{for } k \text{ in range}(1, K+1)\ ]\ ). \tag{12}$$

## 2.4 Architecture of MemoryFormer

We follow the generic design paradigm of the standard transformer architecture [25] using N stacked blocks to build the MemoryFormer. The right part of Figure 3 depicts one building block.

**Multi-Head Attention**  Given an input sequence $\mathbf{X} = (\mathbf{x}_1, \mathbf{x}_2, \cdots, \mathbf{x}_s)^\top \in \mathbb{R}^{s \times d}$, a Norm($\cdot$) layer first normalizes the input. There Memory Layers transform the normalized $\mathbf{X}$ into $\mathbf{Q}, \mathbf{K}, \mathbf{V} \in \mathbb{R}^{s \times d}$, respectively. The tokens in $\mathbf{Q}, \mathbf{K}, \mathbf{V}$ are then evenly split into multiple sub-vectors for multi-head purpose. The calculation of multi-head attention remains untouched as in [25]. Therefore, any other efficiet self-attention techniques such as Flash Attention [10], Linear Attention [16] and KV-Cache can be seamlessly incorporated into MemoryFormer to further increase the forward-time efficiency.

**Memory Block**  In MemoryFormer we use the Memory Block to replace the Feed-Forward Network used in standard transformers. The Memory Block is composed of 2 consecutive Memory Layers, each of which is preceded by a Norm($\cdot$) layer. The norm layer is vital by setting a zero-mean distribution for the input embedding before the hashing operation, and thus the sign function in Eq. (3)

can generate -1 and +1 evenly. Therefore, the output of Eq. (4) can have an uniform distribution so that every bucket in the hash table will be retrieved with equiprobability.

Another detail about the Memory Block is that we omit the intermediate activation function (*e.g.* ReLU, GELU). The hashing operation is a non-linear operation itself. An extra nonlinear function is thus redundant. We have verified through experiments that discarding the non-linear function between the two Memory Layers has no effect on the performance.

In a traditional FFN module, in order to increase the model capacity and performance, the dimensionality of the output token embeddings of the first FC layer is expanded by 4 times, and then restored to hidden size $d$ by the second FC layer. To keep aligned with this design pattern, we set the output dimensionality of the first Memory Layer to be $(\tau + 2) \cdot K$. Remeber that $d = \tau K$, that is, the size of each hash table is $\mathbf{T}_k^1 \in \mathbb{R}^{2^\tau \times (\tau+2) \cdot K}$ in the first Memory Layer of the Memory Block. Therefore, the bit width of $K$ sub-vectors $\mathbf{z}_k$ in the second Memory Layer is 2 bits larger than that of the sub-vectors in the first layer. The size of hash tables in the second layer is $\mathbf{T}_k^2 \in \mathbb{R}^{2^{(\tau+2)} \times d}$, which leads to a capacity 4 times larger than the first layer while restores the dimensionality of the output embeddings back to $d$.

**Computational Complexity.**     So far, the above computing process of one MemoryFormer block is formulated as follows:

$$
\begin{aligned}
\mathbf{X} &= \text{Norm}(\mathbf{X}), \\
\mathbf{Q} &= \text{MemoryLayer}_Q(\mathbf{X}), \quad \mathbf{K} = \text{MemoryLayer}_K(\mathbf{X}), \quad \mathbf{V} = \text{MemoryLayer}_V(\mathbf{X}), \\
\mathbf{Z} &= \mathbf{X} + \text{MultiHeadAttention}(\mathbf{Q}, \mathbf{K}, \mathbf{V}), \\
\mathbf{Y} &= \mathbf{Z} + \text{MemoryLayer}_2(\text{Norm}(\text{MemoryLayer}_1(\text{Norm}(\mathbf{X})))).
\end{aligned} \tag{13}
$$

The amount of floating-point computation of a standard transformer block is $2s^2 d + 12 s d^2$, while the amount of computation of a MemoryFormer block is only about $2s^2 d + \frac{6}{\tau} s d^2 = 2s^2 d + 6Ksd$. The computations originating from FC layers in standard transformer are eliminated by an order of magnitude. The absolute majority of the computational workload now comes from the MultiHeadAttention.

## 3  Experiment

In this section, we conduct thorough experiments on multiple NLP benchmarks to validate the efficiency and effectiveness of the MemoryFormer across different scales. We also compare our model with existing efficient transformer methods.

Given the fact that most large language models only open-source the checkpoint without providing detailed training information, reproducing them is unachievable. Therefore, we employ Pythia [3], a well-developed LLM training framework with completely available dataset and detailed model hyper-parameters, to implement our method. As for training data, the Pile [12] dataset contains 825 GiB corpus with 22 diverse high-quality subsets, which either pre-exists or is constructed from professional and academic fields. We use exactly the same optimizer, scheduler and other hyper-parameters following the setting of Pythia to conduct fair comparisons.

We choose six widely-used evaluation task for our approach: PIQA [4], WinoGrande [23], WSC [24], ARC-E, ARC-C [9], and LogiQA[19]. These tasks range from knowledge to reasoning, forming a comprehensive benchmark for evaluating the all-round capability of large language models.

### 3.1  Evaluation Across Different Scales

We choose Pythia-70M, Pythia-160M, Pythia-410M as the baseline models upon which we build our MemoryFormers. Specifically, MemoryFormer-tiny has the same hidden size and number of layers as Phythia-70M, MemoryFormer-small has the same hidden size and number of layers as Phythia-160M and MemoryFormer-base has the same hidden size and number of layers as Phythia-410M. As for the hyper-parameter of Memory Layer, we fix the value of $\tau$ to be 8, while the number of hash tables $K$ is 64, 96 and 128 respectively for MemoryFormer-tiny, -small and -base model. Notably, considering the sparsity of gradients of the hash tables, we set the learning rate to be 3 times of the baseline learning rate used by the corresponding Pythia model. This is ablated in Sec. 3.3. It's worth noting that, the only one fully-connected layer in the MemoryFormer is the classifier head.

Tab. 1 reports both the computational complexity and the evaluation results of our models compared to baseline. The FLOPs is calculated for one transformer block with the input sequence length of

Table 1: Zero-shot evaluation results on public NLP benchmarks. We use "MF" as the abbreviation for MemoryFormer. "Attn." refers to the computation of $\sigma(\mathbf{Q}\mathbf{K}^\top)\mathbf{V}$. Inference FLOPs are measured for one block with sequence length of 2048.

| Model | Pythia-70M | MF-tiny | Pythia-160M | MF-small | Pythia-410M | MF-base |
|---|---|---|---|---|---|---|
| Layers | 6 | 6 | 12 | 12 | 24 | 24 |
| Hidden Size | 512 | 512 | 768 | 768 | 1024 | 1024 |
| FLOPs w/o Attn. | 6.4 G | 0.4 G | 14.5 G | 1.0 G | 25.8 G | 1.6 G |
| Total FLOPs | 10.7 G | 4.7 G | 20.9 G | 7.4 G | 34.4 G | 10.2 G |
| PIQA | 0.585 | 0.602 | 0.618 | 0.642 | 0.675 | 0.698 |
| WinoGrande | 0.511 | 0.522 | 0.497 | 0.523 | 0.534 | 0.546 |
| WSC | 0.365 | 0.375 | 0.365 | 0.394 | 0.471 | 0.385 |
| ARC-E | 0.380 | 0.437 | 0.440 | 0.461 | 0.517 | 0.585 |
| ARC-C | 0.177 | 0.228 | 0.201 | 0.247 | 0.202 | 0.259 |
| LogiQA | 0.232 | 0.260 | 0.210 | 0.272 | 0.209 | 0.272 |
| Avg. | 0.375 | 0.404 | 0.389 | 0.423 | 0.435 | 0.458 |

Table 2: Comparison of different efficient transformer methods based on Pythia-410M. Inference FLOPs are measured for one block with sequence length of 2048.

| Model | Pythia-410M | Linformer | cosFormer | Performer | MemoryFormer-base |
|---|---|---|---|---|---|
| FLOPs | 34.4 G | 26.1G | 30.0 G | 26.7 | 10.2 G |
| PIQA | 0.675 | 0.527 | 0.522 | 0.643 | 0.698 |
| WinoGrande | 0.534 | 0.511 | 0.506 | 0.496 | 0.546 |
| WSC | 0.471 | 0.635 | 0.605 | 0.433 | 0.385 |
| ARC-E | 0.517 | 0.265 | 0.267 | 0.470 | 0.585 |
| ARC-C | 0.202 | 0.244 | 0.263 | 0.231 | 0.259 |
| LogiQA | 0.209 | 0.207 | 0.264 | 0.236 | 0.272 |
| Avg. | 0.435 | 0.398 | 0.405 | 0.418 | 0.458 |

2048. The experiment results show that MemoryFormer has the minimum computation except for the necessary computation of self-attention. Across all three different model sizes, we achieve better average accuracy on the benchmark than the Pythia baseline. This suggests that the proposed method is able to greatly reduce the computational complexity without compromising the performance.

### 3.2 Comparison with Efficient Transformers

We also compare our models with existing efficient transformer methods to show the superiority of MemoryFormer in both performance and efficiency. We choose Pythia-410M as the baseline, and replace the multi-head attention module of Pythia with the ones proposed by Linformer [26], Cosformer [22], and Performer [8], respectively. Tab. 2 demonstratse the experiment results on the benchmark and the inference FLOPs of each model. We measure the FLOPs using one transformer block with the sequence length of 2048. As shown in Tab. 2, these efficient attention methods can obtain FLOPs-reduction but with considerable performance degradation, while MemoryFormer significantly eliminates the computations and gains better performance. On the other hand, even though existing efficient transformer methods can reduce the computation cost of the self-attention operation, yet we can observe from Tab. 2 that the majority of the computation originates from the fully-connected layers in both the MHA and FFN module as discussed previously. The linear projection operation accounts for the biggest part of the workload when the sequence length is not extremely large in most of the practical scenarios. Utilizing Memory Layer in the embedding space to replace FC layers does provide a new solution to minimize the FLOPs of LLMs.

### 3.3 Ablation Study

**Tradeoff between $\tau$ and $K$.** In Eq. (6) we split an input embedding $\mathbf{x}$ in to K sub-vectors to avoid the explosive growth of memory usage of the hash tables. We study the model performance with different $(\tau, K)$ combination by controlling the model hidden size to be the same. We report the

Table 3: Ablation study on different $\tau$ and $K$. Memory Size refer to the storage space required by the Memory Layer Q.

| $d$ | $\tau$ | $K$ | Val. PPL↓ | FLOPs | Memory Size |
|---|---|---|---|---|---|
| 512 | 4 | 128 | 19.01 | 0.14 G | 2.1 MB |
| 512 | 8 | 64 | 18.82 | 0.07 G | 16.8 MB |
| 510 | 10 | 51 | 18.67 | 0.06 G | 53.5 MB |

Table 4: Val. PPL at 8000 training steps with various LR.

| LR | Val. PPL ↓ |
|---|---|
| 1e-3 | 19.86 |
| 2e-3 | 19.07 |
| 3e-3 | 18.82 |
| 4e-3 | 18.84 |

Table 5: Different expanding bits of Memory Block. #Expanding Bit$= \tau' - \tau$ denotes the number of extra bit of $\mathbf{z}_k$ after expansion. Memory Size denotes the storage space required by Memory Block.

| #Expanding Bit | $\tau'$ | Val. PPL ↓ | Size of Hash Tables | | Memory Size |
|---|---|---|---|---|---|
| 0 | 8 | 19.89 | $\mathbf{T}_k^{M1} \in \mathbb{R}^{256 \times 512}$, | $\mathbf{T}_k^{M2} \in \mathbb{R}^{256 \times 512}$ | 33.6 MB |
| 1 | 9 | 19.26 | $\mathbf{T}_k^{M1} \in \mathbb{R}^{256 \times 576}$, | $\mathbf{T}_k^{M2} \in \mathbb{R}^{512 \times 512}$ | 52.4 MB |
| 2 | 10 | 18.82 | $\mathbf{T}_k^{M1} \in \mathbb{R}^{256 \times 640}$, | $\mathbf{T}_k^{M2} \in \mathbb{R}^{1024 \times 512}$ | 88.1 MB |
| 3 | 11 | 18.54 | $\mathbf{T}_k^{M1} \in \mathbb{R}^{256 \times 704}$, | $\mathbf{T}_k^{M2} \in \mathbb{R}^{2048 \times 512}$ | 157.3 MB |

experiment results in Tab. 3 along with the computation in FLOPs of the corresponding Memory Layer with sequence length of 2048. As is shown in Tab. 3, as the bit width of $\mathbf{z}$ increases, the model performance continues to grow due to the exponentially enlarged capacity of hash tables. However, the memory consumption of the Memory Layer also increases drastically and its computational complexity soon reaches the lower bound. To trade off between the efficiency and memory usage, we conjecture that $\tau = 8$ is a good option for MemoryFormer.

**Larger learning rate.** From Eq. (12) we can observe that the gradients of the hash table are sparse during backward propagation. Some buckets in the hash table might not get updated in one training step. We conjecture that larger learning rate will help remedy the lack-of-gradients situation. We train a MemoryFormer-tiny for 8000 steps with different LR and report the PPL of validation set in Tab. 4. We use initial LR=1e-3 as the baseline learning rate following the settings of Pythia-70M. The best performance is achieved with the learning rate 3 times of the baseline.

**Expanding bit in the Memory Block.** As mentioned in Sec. 2.4, in order to increase the model capacity, we enlarge the dimensionality of the output embedding of the first layer of the Memory Block, which consequently expands the bit-width of the the sub-vector $\mathbf{z}_k$ in the second layer of the Memory Block. We use MemoryFormer-tiny with hidden size $d = 512$, bit-width $\tau = 8$ and number of hash tables $K = 64$ as the baseline model, and report the perplexity results of using different number of expanding bits after 8000 training steps in Tab. 5. We use $\mathbf{T}_k^{M1}$ to denote the hash tables of the 1st layer of the Memory Block, $\mathbf{T}_k^{M2}$ to denote the hash tables of the 2nd layer, and use $\tau'$ to denote the bit-width of $\mathbf{z}_k$ in the second layer after expansion. As shown in Tab. 5, the validation PPL continuously decreases as $\tau'$ gets larger. However, the memory space consumed by the hash tables keeps increasing exponentially. Therefore, we choose 2 as the number of expanding bit in the Memory Block for the trade-off between the space complexity and performance.

**Removing non-linearity in the Memory Block.** We also mentioned in Sec. 2.4 that all activation functions are discarded in the MemoryFormer. We trained a MemoryFormer-tiny on PILE dataset, where we insert a GeLU layer between the two consecutive Memory Layers of the Memory Block for each building block, and report the testing scores on multiple tasks. As shown in Tab. 6, adding an extra GeLU into the Memory Block leads to almost identical result to the baseline MemoryFormer.

### 3.4 Visualization

**Distribution of Hash Bucket.** In a Memory Layer, we do not want most of the sub-vector $\mathbf{z}_k$ hashed to a few popular buckets while the rest of buckets in the table are rarely selected. We expect $\mathbf{z}_k$ to be hashed to all the buckets with the same probability. If so, the output space of the Memory Layer would be diversified enough to enlarge the capacity of MemoryFormer. We use 2048 sequences of 1024 token length to visualize the distribution of the frequency that each bucket is retrieved within a hash table in Figure 4. Specially, we choose the first table $\mathbf{T}_1$ and the last table $\mathbf{T}_{64}$ of the Q,

Table 6: Ablation study on whether to use the non-linearity in the Memory Block.

| Model | PIQA | WinoGrande | WSC | ARC-E | ARC-C | LogiQA | Avg. |
|---|---|---|---|---|---|---|---|
| MemoryFormer-tiny | 0.602 | 0.522 | 0.375 | 0.437 | 0.228 | 0.26 | 0.404 |
| MemoryFormer-tiny + GeLU | 0.595 | 0.522 | 0.375 | 0.441 | 0.211 | 0.265 | 0.402 |

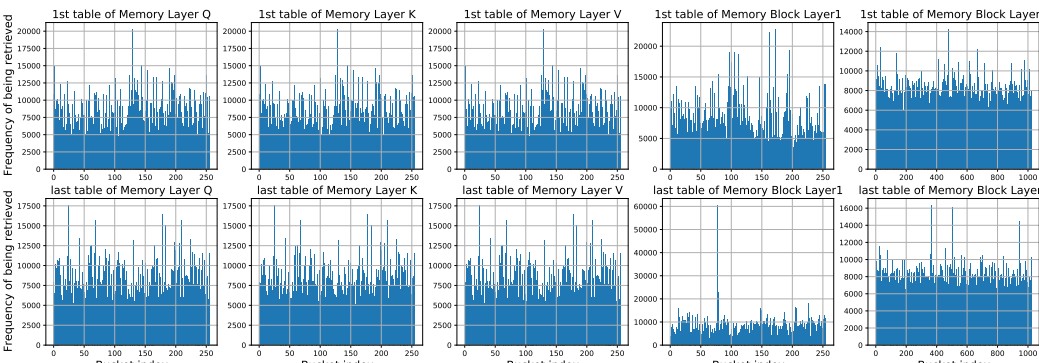

Figure 4: The frequency at which each bucket in the hash table is retrieved.

K, V projection layers and the two layers in the FFN module from the first building block of the MemoryFormer-tiny model. As shown in Figure 4, the number of times each bucket is hashed to by $\mathbf{z}$ is generally uniform.

## 4    Related Works

**Locality Sensitive Hashing.**   Locality sensitive hashing [5, 11, 1] (LSH) is a special kind of hash function which is designed to maximize the collision probability for similar input items. It's wildly adopted in deep learning-based applications. Large scale image retrieval system [18, 15, 2, 30] uses LSH to locate the similar images. Reformer[17] and YOSO [28] both use LSH algorithm to reduce the memory consumption and improve the computational efficiency of the self-attention module. LookupFFN [29] adopt the LSH to accelerate the inference speed of the Feed-Forward Network. SLIDE [7] and MONGOOSE [6] improve the converging speed of neural network training process with the help of locality sensitive hashing algorithms.

**Efficient Transformers.**   Minimizing the computational complexity of the transformer model has always been a center task for the deep learning community. Many previous works are dedicated to reducing the complexity of multi-head attention module to sub-quadratic, such as CosFormer [22], PerFormer [8], LinFormer [26] and so on. [14] uses a sliding window to constrain the attention map within a local range. Besides, there are some researches [13, 20, 27] exploiting the sparsity of the intermediate activation in the FFN (MLP) module to reduce the computation. Recently, with the development of large language models, practical engineering method like FlashAttention [10] brings substantial optimization for the self-attention mechanism.

## 5    Conclusion

In this work, we propose MemoryFormer, a novel transformer architecture that significantly reduces the computational complexity (FLOPs) of transformer model from a new perspective. Unlike existing methods that opt to optimize the computation of the multi-head attention operation, this work provides a new solution from a new perspective. We observe that, in most scenarios the vast majority of computations originates from the fully-connected layers in the transformer model. Thus we focus on removing them. The MemoryFormeruses Memory Layer, which uses locality-sensitive hashing algorithm to perform feature transformation in the embedding space, to replace all the FC layers. It achieves a similar function as the computation-heavy matrix multiplication operation but with much less FLOPs. We successfully eliminate nearly all the computations of the transformer model except for the necessary ones required by the self-attention operation. We validate the efficiency and the effectiveness of MemoryFormer via extensive experiments on public NLP benchmarks.

# Acknowledgement

This work is supported by the National Key R&D Program of China under Grant No.2022ZD0160300 and the National Natural Science Foundation of China under Grant No.62276007. We gratefully acknowledge the support of MindSpore, CANN and Ascend AI Processor used in this research.

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
