# OpenReview forum: "MemoryFormer : Minimize Transformer Computation by Removing Fully-Connected Layers"
_NeurIPS.cc/2024/Conference — NeurIPS 2024 poster_

### Official Review · Reviewer_eumW · 2024-07-03

**Soundness:** 3
**Presentation:** 3
**Contribution:** 4
**Rating:** 7
**Confidence:** 4

**Summary:**

In this paper, the authors present a new transformer model named MemoryFormer, which utilizes locality-sensitive hashing to replace the matrix multiplication operation of the fully-connected layer. This paper can minimize the computational complexity of transformer by removing all the computations except for self-attention. The idea is to use hash tables to store pre-computed features in the memory and retrieve them during inference. The authors evaluate the MemoryFormer on six tasks and compare with other efficient transformer models to show the superiority in saving computations.

**Strengths:**

This paper is well-written and easy to follow. Existing works that try to make the transformer model computationally efficient mainly focus on the self-attention module and FFN module, while this paper tries to address this issue by focusing on the most fundamental part of transformer, the fully-connected layer. This makes the paper stand out for its novelty. The authors use locality-sensitive hashing to simulate the linear projection and use gradient descent via backpropagation to learn the hash tables where the embeddings are stored. The leverage of the memory resources rather than massive parallel matrix multiplication makes the presented model potentially feasible for CPU deployment and inference.

**Weaknesses:**

1. Although the authors experiment on three different model sizes, but they do not provide the exact number of parameters of these models. It would be better if the authors could provide the parameter number.
2. Missing the ablation study about the non-linearity in the FFN module.
3. It would be better to provide some discussion about the training cost given that frequently hashing and retrieving would cost much VRAM I/O of GPU for training.

**Questions:**

1. What is the training cost of MemoryFormer? Is it fast to train such models?
2. Can MemoryFormer be scaled up to reach billions of parameters like Llama-7B?

**Limitations:**

See the weaknesses and questions.

---

> ### Author Rebuttal · Authors · 2024-08-07
>
> ### **1. Data regarding the number of parameters.**
> We report the number of parameters for each model in the row "Model Size" in table below.
> ||Pythia-70M|MF-tiny|Pythia-160M|MF-small|Pythia-410M|MF-base|pythia-1B|pythia-2.8B|pythia-6.9B|
> |-|-|-|-|-|-|-|-|-|-|
> |Layers/Hidden_dim|6/512|6/512|12/768|12/768|24/1024|24/1024|16/2048|32/2560|32/4096|
> |__Model Size__|70M|460M|160M|1.9B|410M|6.7B|1B|2.8B|6.9B|
> |FLOPs w/o Attn.|6.4G|0.4G|14.5G|1.0G|25.8G|1.6G|103.1G|161.1G|412.3G|
> |Total FLOPs|10.7G|4.7G|20.9G|7.4G|34.4G|10.2G|120.3G|182.5G|446.7G|
>
> **It is worth noting that**, in a traditional transformer that is based on fully-connected layers, all parameters are involved in the matrix-multiplication.
> However, the Memory Layer in the MemoryFormer is a sparsely activated module. Only a small fraction ($\frac{1}{2^\tau}$， usually $\frac{1}{256}$) of parameters stored in the hash tables are retrieved for computation during inference while most of the parameters are not used. Therefore, the model size of a MemoryFormer is not the real number of parameters involved in the forward pass. The proposed FLOPs-reduction method is achieved via using more memory space.
>
> ### **2. Ablation study about the non-linearity in the Memory Block.**
> Following the experiment setup of training the MemoryFormer-tiny model, we insert a GeLU layer between the two consecutive Memory Layers of the Memory Block for each building block of the MemoryFormer-tiny. We report the testing scores on multiple tasks after training this model on PILE dataset.
>
> |Model|PIQA|WinoGrande|WSC|ARC-E|ARC-C|LogiQA|Avg.|
> |-|-|-|-|-|-|-|-|
> |MemoryFormer-tiny|0.602|0.522|0.375|0.437|0.228|0.26|0.404|
> |MemoryFormer-tiny + GeLU|0.595|0.522|0.375|0.441|0.211|0.265|0.402|
>
> As shown in the table, adding an extra GeLU into the Memory Block leads to almost identical result to (or even slightly worse than) the baseline MemoryFormer. We will add this experiment into the ablation study.
>
> ### **3. Training cost of the MemoryFormer.**
>
> We report the training speed for MemoryFormers:
> ||MemoryFormer-tiny|MemoryFormer-small|MF-base|
> |-|-|-|-|
> |Training speed (sec/iter)|1.98|4.11|6.60|
>
> Given the fact that GPU is a computationally intensive device with moderate memory i/o bandwidth, and we use custom CUDA kernel that is unable to fully exploit the GPU performance for hashing operation, the training throughput of MemoryFormer is still acceptable. For example, using 2 servers with 8 A100 GPUs, MemoryFormer-small finishs training for 143000 steps on PILE dataset in 6.8 days. The training speed could be faster if our custom CUDA kernel is professionally tuned and optimized.
>
> ### **4. Scalability of the MemoryFormer.**
> A series of experiments on different model sizes (-tiny/-small/-base) proved the scalability of MemoryFormer. As for larger-sized model, We are currently training a MemoryFormer-large based on Pythia-1.4B baseline, which has 24 layers and the hidden dimension of 2048. The trend of the training loss curve of such a model is decreasing normally, which further verifies the scalability.

---

> > ### Comment · Reviewer_eumW · 2024-08-10
> >
> > Thank the authors for your rebuttals carefully. My concerns are well addressed. Thus, I keep my original rate to accept this paper.

---

> > > ### Author Response · Authors · 2024-08-12
> > >
> > > We thank the reviewer for their thorough and insightful review on this work! If you have any further concerns about our work, please leave a comment and we will response in time.

---

### Official Review · Reviewer_vbej · 2024-07-04

**Soundness:** 3
**Presentation:** 3
**Contribution:** 3
**Rating:** 5
**Confidence:** 3

**Summary:**

A FLOPs-reduction strategy designed for transformer is proposed by this paper. The author introduces MemoryFormer, a transformer model that is built using the Memory Layer instead of fully-connected layer. According to the paper, the author claims that the MemoryFormer has the minimum computation because the Memory Layer uses a locality-sensitive hashing algorithm with low computational complexity to compute feature projection instead of using matrix multiplication performed by Fully-Connected layer. The author also designs a differentiable mechanism for the hash tables to be updated, therefore making the MemoryFormer end-to-end trainable like baseline transformer. Besides, the author keeps the necessary computation of self-attention operation untouched, which makes this work orthogonal to other existing efficient transformer designs. Extensive experiments show that the proposed MemoryFormer performs better than other transformer baselines on 6 tasks when built with the same hyper-parameters, such as number of layers and hidden sizes.

**Strengths:**

• The heavy computations of decoder model hinders the further development for LLMs. This paper aims at solving this main issue that the LLMs are currently facing. Exploiting storage to reduce inference computation is a fancy idea.
• The proposed hashing-based Memory Layer requires much less computation than Fully-Connected layer in forward pass. The back-propagation mechanism of the hash tables proposed in Sec.2.2 sounds reasonable.
• The experiment section are comprehensive. The results show that MemoryFormer has better performance while consuming much fewer FLOPs.
• The content of paper is well-written and well-organized. The equations are clear and easy to understand.

**Weaknesses:**

• The proposed method could be further tested with other attention method, for example, incorporating linear attention mechanism into MemoryFormer. However, such experiment is missing.
• The author didn't report the parameter size of the proposed Memory Layer against the corresponding Fully-Connected layer, and did not report the number of parameters of the MemoryFormer model.
• Lack of the experiment for the activation function of Memory Block.

**Questions:**

Since current LLMs require scaling up in model size for more intelligence, I wonder if the author experimented on a larger size of the proposed model? Scalability is crucial.

**Limitations:**

This paper proposed an efficient operation for transformer model, which is experimented via generative decoder model. I’m concerned about the scalability of the proposed method.

---

> ### Author Rebuttal · Authors · 2024-08-07
>
> ### **1. Combining MemoryFormer with other efficient attention method.**
> We incorporate the efficient attention modules proposed by Linformer and cosFormer with MemoryFormer-tiny, and train each model for 8000 steps on PILE dataset.
> |Attn. Type|baseline MHA|Linformer|cosFormer|
> |-|-|-|-|
> |**Val ppl.**|18.82|18.91|18.86|
>
> The loss curves of 3 models before 8000 steps are similar. This confirms the fact that the proposed MemoryFormer is orthogonal to the type of attention module and can works well with them.
> While the validation perplexity of the baseline attention model is indeed lower after 8000 training steps, the impact of linear attention on the MemoryFormer should be reflected by the testing scores on multiple tasks after full training. Due to the limited time for rebuttal, we're unable to fully train these models. We'll add this experiment to the final version.
>
>
>
> ### **2. The parameter size of the proposed Memory Layer and MemoryFormer against fully-connected layer baseline.**
>
> We report the parameter size for Memory Layer and the corresponding Fully-Connected layer of the same hidden dimension. We assume the output dimension is the same as input. We also report the FLOPs required by the two kinds of layers to process a sequence of 2048 tokens.
>
> |Hidden dim|512|768|1024|
> |-|-|-|-|
> |Parameter size of Memory Layer|8.39M|18.87M|33.55M|
> |Parameter size of FC Layer|0.26M|0.59M|1.05M|
> |FLOPs of Memory Layer|0.07G|0.15G|0.27G|
> |FLOPs of FC Layer|0.54G|1.21G|2.15G|
>
> The number of parameters of the MemoryFormer models are reported in the row "Model Size" of the table below:
> ||Pythia-70M|MF-tiny|Pythia-160M|MF-small|Pythia-410M|MF-base|pythia-1B|pythia-2.8B|pythia-6.9B|
> |-|-|-|-|-|-|-|-|-|-|
> |Layers/Hidden_dim|6/512|6/512|12/768|12/768|24/1024|24/1024|16/2048|32/2560|32/4096|
> |__Model Size__|70M|460M|160M|1.9B|410M|6.7B|1B|2.8B|6.9B|
> |FLOPs w/o Attn.|6.4G|0.4G|14.5G|1.0G|25.8G|1.6G|103.1G|161.1G|412.3G|
> |Total FLOPs|10.7G|4.7G|20.9G|7.4G|34.4G|10.2G|120.3G|182.5G|446.7G|
>
> **It is worth noting that**, in the traditional transformer that is based on fully-connected layers, all parameters are involved in the matrix-multiplication.
> However, the Memory Layer in the MemoryFormer is a sparsely activated module. Only a small fraction ($\frac{1}{2^\tau}$, usually $\frac{1}{256}$) of vectors stored in the hash tables are retrieved for computation during inference while most of the vectors are not used. Therefore, the model size of a MemoryFormer is not the real number of parameters involved in the forward pass. The proposed FLOPs-reduction method is achieved via using more memory space.
>
>
> ### **3. Experiment for the activation function.**
> Following the experiment setup of training the MemoryFormer-tiny model, we insert a GeLU layer between the two consecutive Memory Layers of the Memory Block for each building block of the MemoryFormer-tiny. We report the testing scores on multiple tasks after training this model on PILE dataset.
>
> |Model|PIQA|WinoGrande|WSC|ARC-E|ARC-C|LogiQA|Avg.|
> |-|-|-|-|-|-|-|-|
> |MemoryFormer-tiny|0.602|0.522|0.375|0.437|0.228|0.26|0.404|
> |MemoryFormer-tiny + GeLU|0.595|0.522|0.375|0.441|0.211|0.265|0.402|
>
> As shown in the table, adding an extra GeLU into the Memory Block leads to almost identical result to (or even slightly worse than) the baseline MemoryFormer. We will add this experiment into the ablation study.
>
> ### **4. Scalability of the MemoryFormer.**
> The scalability is important for large models to develop emergent abilities. A series of experiments on different model sizes (-tiny/-small/-base) proved the basic scalability of MemoryFormer. As for larger-sized model, We are currently training a MemoryFormer-large based on Pythia-1.4B baseline, which has 24 layers and the hidden dimension of 2048. The trend of the training loss curve is decreasing normally, which further verifies the scalability.

---

### Official Review · Reviewer_vj2X · 2024-07-11

**Soundness:** 3
**Presentation:** 4
**Contribution:** 3
**Rating:** 6
**Confidence:** 4

**Summary:**

This paper introduces a novel neural network call MemoryFormer. This is a modified version for transformer that eliminates the dense layer (FC layer). The proposed MemoryFormer tries to address the problem of high computational complexity of decoder-style generative models. Concretely, the author proposes the Memory Layer which uses locality sensitive hashing (LSH) to achieve the embedding projection which is done by the fully-connected layer in normal transformer. This way, the MemoryFormer eliminates the computational cost of linear projections and keeps only the computational cost of multi-head attention. The author conducts sufficient experiments at different model scale on different NLP benchmarks. The experiment results show that MemoryFormer has better performance than other transformer models with the same #layers and hidden dimension while obviously reduce the FLOPs. The method in this paper shows a different but possible way of building future large language models.

**Strengths:**

1 This paper proposes an interesting way to reduce the computational complexity of transformer neural network. The proposed Memory Layer uses LSH algorithm as a replacement for the linear projections of linear layer. It’s novel to use the memory space to store feature vectors instead of compute them on the fly.
2 The experiment results demonstrate that the proposed model has better performance than baseline across different sizes, showcasing the scalability of the proposed model.
3 The alternative plan for FC layer in transformer is underexplored as it is the most basic building component in neural networks.

**Weaknesses:**

1 This paper doesn’t have any data regarding the inference latency of the proposed MemoryFormer neither on CPU or GPU. The purpose of this paper is reducing FLOPs of transformer, I think such experiment is important.
2 The baseline model Pythia[1] uses 8 benchmarks to evaluate Pythia models while this paper uses 6 benchmarks. This is not a big deal but I hope the author can report the results on the other two.
3 The formulation of Eq.10 might be a little bit confusing if not reading carefully and could be optimized for readability.

[1] Pythia: A suite for analyzing large language models across training and scaling. ICML 2023.

**Questions:**

1 Which model size is being used in the ablation study in Section 3.3? Is it MemoryFormer-tiny, or -small, or base? This paper seems to forget to mention about this information.
2 This paper reports the FLOPs of different models in Tab.1 and 2. What is the inference latency achieved by MemoryFormer?

**Limitations:**

Refer to the weakness.

---

> ### Author Rebuttal · Authors · 2024-08-06
>
> ### **1. Data regarding the inference latency.**
> We measure the inference latency for different models on Intel Xeon 8377C CPU and A100 GPU using custom kernel, and show the results in the table below.
>
> ||Pythia-70M|MF-tiny|Pythia-160M|MF-small|Pythia-410M|MF-base|
> |-|-|-|-|-|-|-|
> |Layers|6|6|12|12|24|24|
> |Hidden Size|512|512|768|768|1024|1024|
> |FLOPs w/o Attn.|6.4G|0.4G|14.5G|1.0G|25.8G|1.6G|
> |Total FLOPs|10.7G|4.7G|20.9G|7.4G|34.4G|10.2G|
> |**CPU latency (ms)**|250.7|200.4|597.6|520.8|1586.3|1581.6|
> |**GPU latency (ms)**|7.2|12.1|15.6|27.6|33.5|64.8|
>
> As shown in the table, the inference of MemoryFormer is faster than Pythia baseline on CPU while MemoryFormer has much less FLOPs than Pythia baseline.
>
> ### **2. Evaluation on more tasks.**
> We further evaluate the models on Lambada and SciQ datasets, and report the complete results as follows.
> ||Pythia-70M|MF-tiny|Pythia-160M|MF-small|Pythia-410M|MF-base|
> |-|-|-|-|-|-|-|
> |Lambada|0.23|0.326|0.398|0.408|0.529|0.524|
> |PIQA|0.585|0.602|0.618|0.649|0.675|0.698|
> |WinoGrande|0.511|0.522|0.497|0.511|0.534|0.546|
> |WSC|0.365|0.375|0.365|0.365|0.471|0.385|
> |ARC-E|0.38|0.437|0.44|0.472|0.517|0.585|
> |ARC-C|0.177|0.228|0.201|0.238|0.202|0.259|
> |SciQ|0.654|0.692|0.774|0.788|0.824|0.811|
> |LogiQA|0.232|0.260|0.210|0.295|0.209|0.272|
> |Avg.|0.392|0.431|0.438|0.466|0.495|0.510|
>
> After evalutaed on the remaining two tasks, the MemoryFormer model still achieves higher average scores than its Pythia baseline which has the same hidden size and number of layers.
>
> ### **3. Improving readability for Eq.10.**
> Thanks for the suggestion. We are considering separating the third line of Eq.10 from the equation, to make the transition of $p(z_k)$ more smooth. We'll revise it in the final version.
>
> ### **4. Which model size is used in the ablation study?**
> We feel sorry we forgot to mention such important information. We use MemoryFormer-tiny to conduct the experiments for ablation study throughout Sec.3.3.

---

### Official Review · Reviewer_omZU · 2024-07-12

**Soundness:** 3
**Presentation:** 3
**Contribution:** 3
**Rating:** 6
**Confidence:** 3

**Summary:**

This work proposes to replace most linear layers of transformers by trainable hash-tables. The new modules---called memory layers---rely on locality sensitive hashing to obtain relevant indices within several hash tables, and returns a linear combination of the associated vectors. To overcome the non-differentiability of the hashing operation, the weights of the linear combination are computed from the inner products between each input vector and its hashed representation. They use memory layers to replace key, query, and value projections, as well as to replace the down and up-projection matrices of the feedforward blocks. Playing on the coarseness of the hashing, this new module can reduce the FLOPs required when compared to a traditional matrix multiplication. They train modified Pythia models and show their approach is outperforming baseline models on multiple reasoning tasks, while using significantly less FLOPs.

**Strengths:**

I find the paper well motivated. Tackling the often dominating cost of the FFW operations is an important research direction. The idea developed in this work is novel and the results are surprisingly good. Improving upon the transformer architecture is not an easy feat and the results suggests that the memoryFormer is better on reasoning tasks while requiring less FLOPs. I find it interesting that the non-linear nature of the hashing operation allows to omit the use of activation functions.

An interesting work overall, but I believe some results/details are missing (see weaknesses).

**Weaknesses:**

To fully evaluate the impact of this work, I am missing some informations:
- What is, in more details, the experimental setup used to trained Pythia models? How many steps are used during training? How are the loss curves for both the Pythia baseline and the memoryFormer? Which hardware was used?
- You should do a comparison in terms of number of parameters. The memoryFormer likely uses an order of magnitude more trainable parameters compared to Pythia models. This makes the comparison between e.g. Pythia-70M and MF-tiny questionable. What scores would you get using a Pythia model with a similar number of parameters?
- Discussion on time complexity are missing from the analysis. I am assuming that the memoryFormer is slower. Speed depends on hardware and I understand that it seems unfair to compare your approach to very optimized matrix-multiplication kernels, but this is still an important limitation to discuss. How many iterations per second during training for a memoryFormer and a Pythia model with the same number of parameters? What is the inference speed? This work seems clearly oriented at proposing a cheaper transformer, and successfully reduces the FLOPs required. Knowing if this translates into speed gains today---or giving ideas on how easy it would be to leverage this gain tomorrow---seems important to evaluate the impact of this work. Given a time budget and a specific machine (cpu or gpu), which size of models would fit the time budget? For those model sizes, would a memoryFormer provide better scores than a baseline Pythia model?
- At the small models scale, I am not sure how trustworthy the scores on reasoning tasks are. What would be the accuracies obtained when answering at random? For instance, WinoGrande is a binary task, and scores are often close to 50%. What are the perplexities reached by the different models?

Overall, you propose an interesting architecture, but I am not entirely convinced by the evaluation. A more faithful account of the implications of using memory blocks on time complexity and on the number of parameters would help.

**Questions:**

See above.

**Limitations:**

See weaknesses. I feel the limitations could have been better discussed in the paper, e.g. time complexity.

---

> ### Author Rebuttal · Authors · 2024-08-06
>
> ### **1.1. What is the experimental setup used to trained Pythia models?  How many steps are used during training?**
> All models are trained for 143000 steps. The total batch size is 1024 samples and the sequence length of each sample is 2048. We use exactly the same experimental setups, such as Adam optimizer, cosine lr decay scheduler, rotary positional embeddings, tokenizer, etc., following the routine done by original Pythia paper. The only difference is that, the learning rate for MemoryFormer is set to be 3 times of the LR used by its corresponding Pythia baseline, which is mentioned in Line.230.
> We will refine the information regarding experimental setup in Sec.3.
>
> ### **1.2. How are the loss curves for both the Pythia baseline and the memoryFormer?**
> We plot the loss curves of training stage for Pythia-410M and MemoryFormer-base model in the rebuttal PDF file. We'll add a more comprehensive figure to our paper in final version.
> ### **1.3. Which hardware was used?**
> We use NVIDIA-A100 server with 8 GPUs for training and testing.
> ### **2. The comparison in terms of number of parameters. What scores would you get using a Pythia model with a similar number of parameters?**
> The aim of MemoryFormer is to reduce the FLOPs of transformer, which is achieved by increasing the number of parameters in exchange for much less computational complexity.
> In the table below, we report the number of parameters in the row of "Model Size". We also show the FLOPs and scores for Pythia models of comparable model sizes.
>
> ||Pythia-70M|MF-tiny|Pythia-160M|MF-small|Pythia-410M|MF-base|pythia-1B|pythia-2.8B|pythia-6.9B|
> |-|-|-|-|-|-|-|-|-|-|
> |Layers/Hidden_dim|6/512|6/512|12/768|12/768|24/1024|24/1024|16/2048|32/2560|32/4096|
> |__Model Size__|70M|460M|160M|1.9B|410M|6.7B|1B|2.8B|6.9B|
> |FLOPs w/o Attn.|6.4G|0.4G|14.5G|1.0G|25.8G|1.6G|103.1G|161.1G|412.3G|
> |Total FLOPs|10.7G|4.7G|20.9G|7.4G|34.4G|10.2G|120.3G|182.5G|446.7G|
> |PIQA|0.585|0.602|0.618|0.642|0.675|0.698|0.700|0.741|0.760|
> |WinoGrande|0.511|0.522|0.497|0.523|0.534|0.546|0.529|0.582|0.631|
> |WSC|0.365|0.375|0.365|0.394|0.471|0.385|0.365|0.385|0.442|
> |ARC-E|0.380|0.437|0.440|0.461|0.517|0.585|0.585|0.635|0.686|
> |ARC-C|0.177|0.228|0.201|0.247|0.202|0.259|0.245|0.301|0.331|
> |LogiQA|0.232|0.260|0.210|0.272|0.209|0.272|0.212|0.214|0.215|
> |Avg.|0.375|0.404|0.389|0.423|0.435|0.458|0.439|0.476|0.511|
>
> **It is worth noting that**, in a traditional transformer that is based on fully-connected layers, all parameters are involved in the matrix-multiplication.
> However, the Memory Layer in the MemoryFormer is a sparsely activated module. Only a small fraction ($\frac{1}{2^\tau}$, usually $\frac{1}{256}$) of vectors stored in the hash tables are retrieved during inference while most of the vectors are not used. Therefore, the model size of a MemoryFormer is not the real number of parameters involved in the forward pass.
> ### **3. Discussion on time complexity.**
> The speed of the proposed method is indeed an important aspect. We report the average training speed for models of similar sizes:
>
> ||MF-tiny|Pythia-410M|MF-base|pythia-6.9B|
> |-|-|-|-|-|
> |Model Size|460M|410M|6.7B|6.9B|
> |Training speed (sec/iter)|1.98|1.96|6.60|6.57|
>
> The inference latency is related to specific hardwares. We show the latency tested on Intel Xeon 8377C CPU and A100 GPU using custom kernel in the table below. We take the average time of feeding a sequence of 2048 tokens into the model for 100 times.
>
> ||MF-tiny|Pythia-410M|MF-base|pythia-6.9B|
> |-|-|-|-|-|
> |Model Size|460M|410M|6.7B|6.9B|
> |CPU Inference (ms)|200.4|1586.3|1581.6|9788.8|
> |GPU Inference (ms)|12.1|33.5|64.8|163.2|
>
> According to the table, the MemoryFormer achieves lower inference latency on CPU and GPU than pythia model of similar number of parameters. This result could be bettered via optimizing the hash kernel. Currently, MemoryFormer's latency on CPU/GPU platform is primarily bounded by memory bandwidth. GPU hardware is a computationally intensive chip, which gives rise to MatMul-based models with large FLOPs. Yet there are other types of computing chips being broadly developed. For example, Processing-in-Memory (PIM) architecture[1] is a frontier ASIC. Such hardwares[2,3,4,5] incorporate the arithmetic units into the design of RAM circuit to provides extremely large bandwidth with massive memory space, but have relatively lower compute capability. The inference paradigm of the proposed MemoryFormer is different from traditional transformer, which indicates that LLMs can be deployed on low-compute-capability platforms while achieving ideal performance and latency.
> ### **4.1 Accuracies obtained when answering at random?**
> We test the accuracies of randomly initialized models:
> ||PIQA|WinoGrande|WSC|ARC-E|ARC-C|LogiQA|Avg.|
> |-|-|-|-|-|-|-|-|
> |Random Pythia|0.509|0.489|0.516|0.251|0.255|0.249|0.378|
> |Random MF|0.501|0.488|0.519|0.263|0.235|0.277|0.381|
>
> PIQA, WinoGrande, WSC are binary tasks, so their results are around 50%. The answers of ARC-E, ARC-C and LogiQA include 4 choices, so their results are around 25%.
> The average score is a reasonable indicator for the performance of the model.
> ### **4.2 What are the perplexities reached by the different models?**
> Perplexities reached by the different models:
> ||Pythia-70M|MF-tiny|Pythia-160M|MF-small|Pythia-410M|MF-base|
> |-|-|-|-|-|-|-|
> |ppl|24.04|19.88|12.68|11.02|9.12|8.94|
>
> MemoryFormer has lower ppl than corresponding baseline.
>
> Reference:\
> [1] A survey on processing-in-memory techniques: Advances and challenges. Elsevier 2023.\
> [2] Computing Utilization Enhancement for Chiplet-based Homogeneous Processing-in-Memory Deep Learning Processors. ACM GLSVLSI 2021.\
> [3] Dojo: The microarchitecture of tesla’s exa-scale computer. IEEE HCS 2022.\
> [4] The Groq Software-defined Scale-out Tensor Streaming Multiprocessor: From chips-to-systems architectural overview. IEEE HCS 2022.\
> [5] Chiplet Cloud: Building AI Supercomputers for Serving Large Generative Language Models. arxiv/2307.02666.

---

> > ### Comment · Reviewer_omZU · 2024-08-11
> >
> > I thank the authors for their detailed rebuttal. My concerns related to training or inference speed have been answered. For a target average score, it seems it is slower to train a MemoryFormer vs a transformer, but there is hope that hardware improvements could address this. Moreover, the main table provided answers my question on which dense model size is needed to have the same performance as a MemoryFormer. While the gap in parameters is quite large, the gain in FLOPs induced by the sparsity is significant. Overall, I believe the novelty of this work is strong enough to be accepted at the conference. I am changing my score to a 6.

---

> > > ### Author Response · Authors · 2024-08-12
> > >
> > > We thank the reviewer for their thorough and insightful review on this work. If you have any further concerns about our work, please leave a comment and we will response in time.

---

### Author Rebuttal · Authors · 2024-08-07

We provide the training loss curves for Pythia-410M and MemoryFormer-base in the rebuttal PDF file.

---

### Public Comment · ~Anastasiia_Filippova2 · 2025-08-01

Dear Authors,

Thank you for an interesting contribution.

I would like to ask how your work relates to [LookupFFN](https://arxiv.org/html/2403.07221v1). In the related work section, you mention that LookupFFN focuses on reducing FLOPs during inference. Forgive me if I’ve overlooked something, but if I understand correctly, the motivation and approach seem quite similar to yours, namely, reducing FLOPs for both inference and training. Both works also seem to address similar challenges, such as computing gradients with respect to the input ( making expressions like y=T[h(x)] differentiable) and managing memory usage by avoiding large table storage through input partitioning.

Could you please clarify how your method differs from LookupFFN in terms of contributions or implementation? I’d appreciate your thoughts on this. Thank you again for the interesting work.

Anastasiia

---

> ### Public Comment · ~Ning_Ding4 · 2025-09-08
>
> Greetings, Anastasiia
>
> This work differs from LookupFFN in the following three aspects:
>
> (1) Operator design level difference. (suppose $x$ is input and $y$ is output)
>
> Forward of LookupFFN still contains a fully-connected layer, which **increases** FLOPs:
>
> $z_k=x R_k , R_k \in \mathcal{R}^{~ d \times \tau} , y = \sum_{k=1}^K [T_k] _{z_k}$.
>
> Forward of MemoryLayer does **NOT** include fully-connected layer, which greatly **reduces** FLOPs:
>
> $z_k$= split($x$, num_chunk = K) = $x$.view(K, x.size(-1)/K), $y=\sum_{k=1}^K [T_k] _{z_k}$.
>
> (2) Architecture design level difference.
>
> LookupFFN is the replacement for the whole FFN module (two fully-connected layers).
> There are multiple FC layers in a block (Q/K/V/output_projection/$R_k$).
> LookupFFN does not change the attention module.
>
> MemoryLayer is the replacement for a single fully-connected layer. There are **NO** FC layer in a block.
> The MemoryFormer change the architecture of the attention module, where the Q/K/V projection are replaced with MemoryLayers and the output_projection is removed.
>
> (3) Experimental setting difference.
>
> LookupFFN is tested with encoder-only model , while MemoryFormer is tested with decoder-only model.
>
> ***
> ***
>
> This work expands the great idea of ​​LookupFFN, who accelerate the inference speed by utilizing LookupFFN (FC layer + LSH + learnable tables) to reduce the FLOPs of FFN module (two FC layers).
>
> MemoryFormer try to reduce the FLOPs of the whole transformer much further. Our practice is that we replaced all FC layers with pure (LSH + learnable tables).
>
> ***
> ***
> I hope my reponse could address your concern.
>
> Ning

---

### Decision · Program_Chairs · 2024-09-25

**Decision:**

Accept (poster)

**Comment:**

This paper introduces a novel transformer architecture that replaces fully-connected layers with memory layers using a locality-sensitive hashing (LSH) approach. This significantly reduces computational complexity (FLOPs) while maintaining model performance.

Strengths:
* Novelty: The paper presents a novel method of reducing FLOPs by replacing linear projections with memory layers, offering an efficient alternative to fully-connected layers, a fundamental part of transformers. The approach is both unique and timely, given the push to make large language models (LLMs) more efficient.
* Performances: MemoryFormer outperforms baseline models like Pythia on several reasoning tasks while requiring significantly fewer FLOPs. The experiments are comprehensive across different model sizes, and the paper demonstrates better scalability than existing methods.

Weaknesses and Limitations:
* Training Complexity: Several reviewers raised concerns about the training speed and hardware dependency. The method, while reducing FLOPs, does not necessarily translate into faster training, particularly when compared to heavily optimized matrix-multiplication-based models.
* Parameter Size: While the paper reduces computational complexity, it does so at the expense of increasing the model size, raising questions about memory efficiency and scalability.
* Limited Evaluation: Some reviewers pointed out the need for additional tasks and metrics in evaluation, including a clearer comparison of parameter sizes and in-depth latency studies, which were partially addressed in the rebuttal but could have been more thorough.

Overall, the paper is technically solid, presents a high-impact novel idea, and provides compelling experimental results. However, the evaluation could benefit from deeper exploration of limitations around speed and scalability. Reviewers generally support the paper’s acceptance, but some remain cautious about the practicality of the proposed method on large-scale models. I recommend acceptance as the new idea may spark further discussions.